# A Novel Approach to the Bioluminescent Detection of the SARS-CoV-2 ORF1ab Gene by Coupling Isothermal RNA Reverse Transcription Amplification with a Digital PCR Approach

**DOI:** 10.3390/ijms22031017

**Published:** 2021-01-20

**Authors:** Zhongjie Fei, Rongbin Wei, Chu Cheng, Pengfeng Xiao

**Affiliations:** State Key Laboratory of Bioelectronics, School of Biological Science and Medical Engineering, Southeast University, Nanjing 210096, China; 230189585@seu.edu.cn (Z.F.); 230179152@seu.edu.cn (R.W.); 230208215@seu.edu.cn (C.C.)

**Keywords:** SARS-CoV-2, accurate and rapid identification, RT-LAMP-BART, digital PCR system, digital RT-LAMP approach

## Abstract

The COVID-19 pandemic caused by the SARS-CoV-2 virus, which first emerged in December 2019, represents an ongoing global public health emergency. Here, we developed an improved and highly sensitive approach to SARS-CoV-2 detection via coupling bioluminescence in real-time (BART) and reverse-transcriptase loop-mediated amplification (RT-LAMP) protocols (RT-LAMP-BART) and was also compatible with a digital LAMP system (Rainsuit), which did not allow for real-time quantification but did, nonetheless, facilitate absolute quantification with a comparable detection limit of 10^4^ copies/mL. Through improving RNA availability in samples to ensure the target RNA present in reaction, we additionally developed a simulated digital RT-LAMP approach using this same principle to enlarge the overall reaction volume and to achieve real-time detection with a limit of detection of 10 copies/mL, and with further improvements in the overall dynamic range of this assay system being achieved through additional optimization.

## 1. Introduction

COVID-19 (coronavirus disease 2019) is a pandemic disease caused by SARS-CoV-2 (severe acute respiratory syndrome coronavirus-2) [1] that represents a serious public health emergency throughout the world. Nucleic acid-based diagnostic testing has played a central role in early patient diagnosis and contact tracing efforts [1,2,3].

Several different approaches to nucleic acid-based SARS-CoV-2 detection have been published to date [4,5,6,7,8,9,10]. For example, CRISPR/Cas-based approaches have been developed that can detect RNA samples containing over 1 × 10^4^–1 × 10^5^ copies/mL (SHERLOCK) or 1 × 10^4^ copies/mL (DETECTR) of viral RNA within 1 h. However, these novel approaches are often relatively complex and/or require access to expensive instrumentation without providing sufficient sensitivity advantages relative to more traditional detection techniques such as RT-LAMP [11]. As such these approaches are generally not convenient or conducive to widespread use.

LAMP assays are rapid, sensitive, and specific [12,13], and are more tolerant to the PCR inhibitors that are commonly detected in clinical samples [14,15,16], reducing the requirement for rigorous sample clean-up and DNA extraction (Figure 1A). The theoretical throughput of such LAMP assays can be enhanced using lab-chips. A given LAMP assay utilizes six specific primers and a strand-displacing Bst DNA polymerase to amplify up to 10^20^ copies of a target DNA sequence within 1 h. In addition to being highly tolerant of assay inhibitors, LAMP assays can be conducted at a single constant temperature (65 °C), eliminating the requirement for complex sample preparation or access to a thermocycler while yielding sensitive, specific, and robust results [17,18,19,20]. We have also previously optimized a LAMP-BART technique by utilizing 2-deoxyadenosine-5-(α-thio)-triphosphate (dATPαS) instead of deoxyadenosine triphosphate (dATP), thereby yielding a constant background signal and eliminating high levels of false-positive signals when conducting this assay [21].

Bioluminescence in real-time (BART)-based techniques, in contrast, enable the convenient, rapid, and sensitive detection of specific DNA sequences without the need for complex instrumentation while offering significant sensitivity advantages over similar reverse-transcriptase loop-mediated amplification (RT-LAMP)-based rapid detection approaches [17,22]. In this study, we explored the utility of an RT-LAMP-BART detection strategy that is capable of detecting the bioluminescent signal generated by LAMP-derived pyrophosphate in real-time, thereby further amplifying the underlying nucleic acid signal to yield an assay that is more sensitive than other similar pyrophospholuminescent assays. (Figure 1B) [23,24] The theoretical amount of inorganic pyrophosphate (PPi) generated via PCR from a single copy of a 50 base pair DNA template over 23 PCR cycles is 8.4 × 10^−14^ mol (2 × 500 × 2^23^/6 × 10^23^ = 8.4 × 10^−14^), which is significantly higher than the pyrophosphate detection limit (1.56 × 10^−15^ mol) [25,26]. LAMP has a significantly higher amplification efficiency relative to PCR, and as such a LAMP-BART approach offers significant sensitivity and efficiency advantages, theoretically enabling the detection of a single copy of template DNA [14,25,26,27].

However, conventional LAMP-BART-based assays have rarely been combined with dPCR technology to date. In this study, optimized LAMP-BART assays were studied and found to offer advantages over traditional LAMP-BART assays, which are challenging to conduct in a microfluidic context owing to assay-specific parameters.

Digital PCR (dPCR) is an absolute quantification approach that is used to determine the amount of starting low-concentration DNA template based upon the Poisson distribution [28]. Such dPCR approaches utilize individual droplets derived from a given assay sample as an individual microassay, allowing for a high-throughput microfluidic-based approach to analyzing a given sample [23]. This approach is also compatible with multiplexing by using different fluorophores or by varying the fluorescent intensity of a given probe [29,30,31]. In the present study, we demonstrate the efficacy of an RT-LAMP-BART-based approach to the amplification and bioluminescent detection of an RNA template in microfluidic droplets. An RNA concentration gradient distributed across a series of droplets was used to demonstrate the dependence of light emission on starting RNA concentration, enabling the user to perform a quantitative, low-volume, high-throughput assay. Overall, in this study, we demonstrate that this RT-LAMP-BART assay can be coupled with a dPCR instrument and a simulated dPCR approach to facilitate the rapid and sensitive detection of the SARS-CoV-2 ORF1ab Gene.

## 2. Results and Discussion

### 2.1. The Relative Feasibility of Using dATPαS Instead of dATP in RT-LAMP Reactions

For the present study, we adapted RT-LAMP assays targeting the SARS-CoV-2 Orf1ab gene as they have previously been shown to be highly sensitive and specific [32].

To test the feasibility of our optimized RT-LAMP approach, tubes containing conventional or optimized RT-LAMP reagents were combined with the target RNA sequence (tubes containing conventional or optimized RT-LAMP-BART reagents without target RNA sequences were used as a negative control), after which the reaction was heat-activated and maintained at a constant temperature of 65 °C for 40 min. In the optimized RT-LAMP, dATP was replaced with dATPαS in the RT-LAMP, whereas the other dNTPs were the normal monomers.

Relative to the conventional reagent mixture, the optimized reagent mixture had the same amplification effect in RT-LAMP (Figure 2A), consistent with our previous findings [33] that dATPαS can eliminate false positive signal and better facilitate amplification than dATP in LAMP attributable to dATP (Figure 2B). Although this bioluminescent signal-based method could be used to detect SARS-CoV-2, it could not achieve real-time and accurate quantification. Therefore, we tried to combine this method with digital PCR, which has hardly been tried before.

### 2.2. Assessment of a Digital RT-LAMP-BART Assay Approach

We next evaluated the feasibility of a droplet-based digital LAMP-BART assay approach by using target RNA (10^6^ copies/mL) together with a conventional or optimized RT-LAMP-BART mixture and the DROPDx-2044HT digital PCR system (Suzhou, Jiangsu, China). This system is able to generate roughly 20,000 stable effective droplets using the Mono Flex droplet generation technology, with a target RNA volume of 0.0005 μL per droplet. The stability of the overall reaction system was ensured by using Droplet Generation Oil and Stabilizer. When this approach was conducted with a conventional RT-LAMP-BART reagent mixture in the DROPDx-2044HT digital PCR system, all droplets generated by this system yielded bioluminescence (Figure 3A). This was consistent with the overall assay principle outlined in Figure 1A, wherein PPi is converted to ATP by ATP sulfurylase, with this ATP subsequently generating a bioluminescent signal through the action of the luciferase enzyme. When ATP is present within the reaction mixture, a bioluminescent signal will thus be generated even in the absence of PPi. As dATP is structurally similar to ATP, it can also drive luciferase-mediated luciferin transformation into oxidized luciferin, producing bioluminescence, as confirmed via pyrosequencing [23,24]. As such, all droplets containing the conventional RT-LAMP-BART reagent mixture yielded bioluminescent signal even when template concentrations were low, making it impossible to distinguish between positive and negative signals (Figure 3A_1_,A_2_). In contrast, when the optimized RT-LAMP-BART reagent mixture was used with the DROPDx-2044HT digital PCR system, some droplets exhibited (Figure 3A) bioluminescence whereas others did not (Figure 3B), enabling us to clearly differentiate between positive and negative signals so as to facilitate the counting thereof. As such, our conventional BART reagent mixture was incompatible with this dPCR approach, whereas our optimized RT-LAMP-BART reagent mixture was amenable to use in this assay context. 

To further assess the accuracy and sensitivity of this quantification approach, we next tested this assay using RNA concentrations between 10^3^ and 10^6^ copies/mL. The number of droplets that yielded a bioluminescent signal in this assay context increased with increasing sample concentration (Figure 4A–C), and this was confirmed by counting the number of positive droplets (Figure 4A_1_–C_1_).

As shown in Table 1, using the number of copies in a given droplet reaction (target RNA concentration × reaction volume) and N (the total number of droplets; 20,000), the number of copies per droplet (λ) could be calculated (Copies per droplet = −ln(Nn/N) = Copies in the reaction/Total droplets). Based upon the Poisson distribution, it was thus possible to calculate the number of theoretical positive and negative droplets (Np and Nn, respectively) for a given assay setup as follows (λ = −ln(Nn/N)). At target RNA concentrations of 10^4^, 10^5^, and 10^6^ copies/mL, theoretical Np values were 10, 99, and 969, respectively. The actual experimental Np values at these concentrations (9, 77, 669) were consistent with these theoretical calculations shown in Table 1 (Figure 4A_1_–C_1_). The actual RNA concentration was calculated using the λ value as follows: λ = −ln(Nn/N) = Reaction Concentration in (in Copies/μL) × droplet volume. Reaction Concentration = −ln(Nn/N)/droplet volume. Furthermore, the observed target RNA concentrations in this assay (0.98, 8.89, 70.06 copies/μL) were consistent with the predicted RNA concentrations (1, 10, 100 copies/μL) across the tested RNA concentration range. When RNA was further diluted to 10^3^ copies/mL, however, the results tended to become inconsistent and often undetectable (data not shown). While this method was able to achieve absolute precise quantification and was very convenient and affordable, it thus had a detection limit of 10^4^ copies/mL (1 copies per reaction), thus offering no advantage over other methods such as the RT-LAMP-BART approach discussed above or CRISPR/Cas-based methods. This was due to an intrinsic limitation of the DROPDx-2044HT digital PCR system wherein the reaction mixture volume was 10 μL, rather than a consequence of a lack of methodological sensitivity. The theoretical detection limit for this reaction system is 10^3^ copies/mL (1 copy per reaction), but taking just 10 μL from a 100 μL RT-LAMP-BART reaction mixture caused the copies present within the reaction to become too inconsistent. When testing actual clinical samples, we generally use about 100 microliters of ultrapure water to dissolve the extracted RNA or to directly dissolve the collected samples for testing.

However, in conventional RT-LAMP, RT-PCR, digital PCR systems, or biosensor-based detection [34], the RNA sample or DNA volume is 2–5 mL, such that when samples contain low concentrations of RNA the actual RNA may not be present in the reaction. Therefore, while many novel approaches to single copy RNA detection have been developed, they still have a detection limit of 10^3^–10^4^ copies/mL. This same limitation constrained the digital RT-LAMP-BART assay in the present study. It is thus vital that RNA availability in low concentration samples be improved without altering RT-LAMP reaction efficiency in order to further improve detection limits. In the RT-LAMP reaction system, blind proportional amplification would lead to unstable amplification reaction and decreased sensitivity, while it is combined with digital PCR technology perfectly solved this problem, through dividing a large RT-LAMP reaction system into smaller ones which could increase the utilization rate of template RNA as much as possible in the form of normal RT-LAMP reaction system. If all the template RNA extracted from the sample (100 μL) were present in the reaction, this would significantly improve the overall detection limit. To overcome this limitation and improve the utilization of sample RNA, we therefore next developed a well-based simulated digital RT-LAMP-BART protocol. Additionally, this droplet-based digital LAMP-BART approach was unable to directly establish the target sample concentration based upon the number of negative signals owing to the uncertainty in the N (total droplet number) value (Figure 4A_1_–C_1_), leading us to explore the development of a simulated digital RT-LAMP assay capable of doing so in Section 2.3.

### 2.3. Assessment of Bioluminescent Signal-Based Quantification in a Simulated Digital RT-LAMP Assay

Our simulated digital RT-LAMP assay protocol was designed using the same principles as were used for the above bioluminescent-based digital RT-LAMP assay, treating individual wells of a 96-well plate as individual droplets. To improve the utilization of RNA in a given sample, the total RT-LAMP-BART reaction mixture was used for the reaction.

To evaluate the accuracy and sensitivity of this simulated digital bioluminescent RT-LAMP assay, we prepared wells containing between 10 and 10^4^ copies/mL of target RNA. As shown in Table 2, we were able to utilize the number of RNA copies per reaction (Target RNA concentration in each reaction × reaction volume) and the number of wells (N = 96) to calculate the number of copies per well (λ = Copies in the reaction/N). We were then able to calculate the number of theoretical positive and negative wells (Np and Nn) based upon the Poisson distribution as follows: λ = −ln(Nn/N). Overall, at RNA concentrations of 10, 10^2^, 10^3^, and 10^4^ copies/mL, Np was predicted to be 1, 10, 62, and 96, respectively. The actual observed values (2, 8, 58, 96) were consistent with these predicted values (Figure 5A–D). The actual RNA concentration per well could then be calculated based upon the Nn value. Based on the equation λ = −ln(Nn/N) = Concentration in reaction × V (volume of each well), the relationship between Nn and RNA concentration (in copies/μL) was defined as: Concentrations in reaction = −ln(Nn/N)/V = −ln(Nn/96)/10.4 = −ln(Nn/96)/10.4 = ln(96/Nn)/10.4.

The calculated RNA concentrations under these four reaction conditions were 0.09, 0.084, 0.885, and 9.6 copies/μL, consistent with the predicted concentrations (Table 2: 0.001, 0.01, 0.1, 1 copies/μL. Overall, our simulated digital LAMP assay thus had a sensitivity of 10 copies/mL (Table 2), making it more sensitive than both our digital LAMP and RT-LAMP-BART assays.

While in the tested format this simulated digital LAMP assay had a quantification range of 10–10^4^ copies/mL of target RNA, making it primarily useful for detecting low-concentration RNA samples. It is a very flexible method, and by adjusting the number of wells (if a machine capable of more parallel wells was designed), it is possible to further expand this precise quantification range (the more wells, the larger precise quantification range). This approach can also improve the overall sensitivity and stability of this assay (if a machine capable of such parallel analysis was designed, this range would expand to 10–(+∞) copies/mL).

Furthermore, we were also able to use this method for simultaneous real-time and endpoint detection of assay results, which could be acquired in between 15–40 min (Figure 5A–A_2_,B–B_2_). Each well of a 96-well plate could be detected in real-time (Figure 5A_1_,B_1_, and by accumulating these signals we were able to establish the overall bioluminescent signal values for 96 wells in real-time (Figure 5A_2_,B_2_). By comparing the peaks (about 30 min) of light intensity, we were also able to define the RNA concentration. In general, this approach can accurately quantify target RNA levels in real-time, yielding results within 20 min which greatly reduced the probability of false positives. It is well known that the longer the RT-LAMP reaction time, the higher the probability of false positive [35].

### 2.4. Assessment of Bioluminescent Signal-Based Quantification in a Simulated Digital RT-LAMP Assay as Verified with a SARS-CoV-2 Orf1ab Visual RT-LAMP Kit

The SARS-CoV-2 Orf1ab Visual RT-LAMP kit was used to verify our simulated digital RT-LAMP assay. Pseudoviral plasmid reference material and negative control samples in the Kit were prepared using a two-step concentration gradient (10^4^, 10^3^ copies/mL) and tested via simulated digital RT-LAMP assay, qPCR, and visual assay. As shown in Figure 6, when the target RNA concentration was 10^4^ copies/mL, the results of all three methods were positive and the qPCR assay took 32 min, whereas the visual RT-LAMP assay required 35 min and the simulated digital RT-LAMP assay took 20 min. When target RNA concentrations were 10^3^ copies/mL (reached the detection limit of the SARS-CoV-2 Orf1ab Visual RT-LAMP kit), qPCR, and visual assay became unsteady (sometimes positive while sometimes negative). It was due to the template was not able to actually present in RT-LAMP every time. When the RNA were actually present in RT-LAMP, all three methods again yielded positive results and required 60, 65, and 20 min, respectively. When negative control samples were tested, all three approaches yielded negative results. The results of a simulated digital RT-LAMP assay conducted using positive control samples from the kit (Table 3) were consistent with the results in Table 2.

### 2.5. The Specificity and Assessments of Bioluminescent Signal-Based Quantification in a Simulated Digital RT-LAMP Assay as Verified with RT-qPCR

As shown in Table 4, in combination with the results of simulated digital RT-LAMP assay and RT-qPCR, we found that digital PCR has good specificity and required less time and cost less. The advantages of this method, which are high sensitivity, quantitative accuracy and speed outweigh the disadvantages of its more complicated protocol and operation. In terms of cost, this method has an advantage over fluorescence detection because it does not require an excitation source. However, because it is combined with digital PCR, the cost is not low. However, as the equipment upgrades, the operation will be simplified, and the cost can be further reduced. In general, this method is of great significance in accurate detection and sequencing of various viruses.

## 3. Materials and Methods

### 3.1. Reagents

A pseudoviral SARS-CoV-2 ORF1ab reference plasmid was synthesized by Sangon Biotech (Shanghai, China), and contained the complete SARS-CoV-2 ORF1ab gene sequence. SARS-CoV-2 Orf1ab Visual RT-LAMP Kits were obtained from Biolab (Beijing, China). Adenosine 5′-phosphosulfate (APS), adenosine 5′-triphosphate (ATP) sulfurylase, Bst polymerase v2.0 Warm Start, and 10× Bst buffer were purchased from NEB (Beijing, China). D-Luciferin sodium was obtained from Promega (Shanghai, China). AMV Reverse Transcriptase (AMV) was purchased from Woosen Biotechnology (Shanghai, China). Droplet generation oil and stabilizer were obtained from Suzhou Rainsure Scientific Co. Ltd (Suzhou, Jiangsu, China). A DROPDx-2044HT digital PCR system was obtained from Suzhou Rainsure Scientific Co., Ltd. An ultra-weak luminescence analyzer was obtained from Jianxinlitou (Beijing, China). The all-in-one BZ-X800E fluorescence microimaging system was obtained from KEYENCE (Shanghai, China).

### 3.2. Template and RNA Gradient Preparation

The SARS-CoV-2 ORF1ab pseudoviral plasmid (target RNA) was used to prepare a 6-step concentration gradient (10^6^, 10^5^, 10^4^, 10^3^, 10^2^, 10 copies/mL) for downstream analyses, and a negative control sample was also prepared (containing reaction mixture without target RNA). Ultrapure water was used as a diluent.

### 3.3. Primer Design

LAMP primers specific for the SARS-CoV-2 ORF1ab gene were obtained from a SARS-CoV-2 Visual RT-LAMP Kit (Biolab, Beijing, China) and were based on the conserved ORF1ab gene sequence published in GenBank (GenBank: NC_045512.2). LAMP primers were designed using LAMP Designer 2.0, and included two internal primers (FIP/BJP), two external primers (F3/B3), and two Loop primers (LoopF/LoopB). For full primer details, see Table 5.

All primer pairs used for RT-LAMP assays were synthesized and HPLC-purified by Shanghai Biotech (Shanghai, China). We additionally confirmed the specificity of these primers and found that they were not prone to primer-dimer formation, tailing, or miscellaneous banding.

### 3.4. The Feasibility of RT-LAMP Assay

The RT- LAMP assays were performed with 500 μL reaction mixtures containing 100 μL of template pseudoviral plasmid reference RNA, 50 μL of Bst polymerase v2.0 Warm Start (8 U/μL), 70 μL of 10× Bst buffer, 50 μL of AMV (5 U/μL), 5 μL of each of the inner primers (FIP and BIP, 2.4 µmol/L), 5 μL of each of the outer primers (F3 and B3, 0.4 µmol/L), 5 μL of each loop primer (LF and LB, 1.2 μmol/L), 1 μL of betaine (0.8 mol/L), and 200 μL of reaction mixture (0.3 mM deoxynucleotide triphosphates [dNTPs]). An optimized RT-LAMP mixture using dATPαS instead of dATP (Wuhu Huaren Science and Technology Co. Ltd., Wuhu, Anhui, China) was used for reactions, while normal monomers were used for all other dNTPs. An optimized RT-LAMP mixture using dATPαS instead of dATP (Wuhu Huaren Science and Technology Co. Ltd., Anhui, China). The reaction was monitored with a thermostat (Omega, UK) at 60 °C for 50 min. The LAMP products (2 μL) were separated with agarose gel electrophoresis and observed in a gel imaging system.

### 3.5. RT-LAMP Bioluminescent Reagent Preparation

RT-LAMP Bioluminescent reactions were conducted in a 1000 μL volume composed of a 50 μL RT-LAMP mixture and a 500 μL bioluminescent mixture.

The 500 μL bioluminescent mixture included 20 mM Tris-acetate (pH 8.0), 2 mM Mg(Ac)_2_, 0.4 mM beetle luciferin potassium salt, 6 μM KCl, 10 mM DTT, 5 μM adenosine-5′-O-phosphosulfate (APS), 1.46 ng/mL luciferase, and 0.2 U/mL ATP sulfurylase.

### 3.6. SARS-CoV-2 ORF1ab Gene Detection

#### 3.6.1. Bioluminescent Signal-Based Digital LAMP

For digital LAMP assays, 10 μL of the new or conventional RT-LAMP-BART reaction mixture, 70 μL Droplet Generation Oil, and 10 μL Stabilizer were added to the microfluidic chip. Droplets were generated using the DROPDx-2044HT digital PCR system (Suzhou Rainsure Scientific Co., Ltd., Suzhou, Jiangsu, China). The microfluidic chip matched with the digital PCR system was incubated for 35 min at 65 °C, after which the system was used to read the bioluminescent signal.

#### 3.6.2. Bioluminescent Signal-Based Simulated Digital LAMP

The total RT-LAMP-BART reaction mixture volume (1000 μL) was equally distributed across 96-well plates (10.4 μL/well) such that each well was equivalent to each droplet generated via the above approach. The plates were then placed in the BZ-X800E system (KEYENCE, Tokyo, Japan), and images and light intensity values were recorded following a 20–40 min incubation at 65 °C. This system utilized a combination of pixel binning and local CCD readings to achieve higher sensitivity.

### 3.7. Simulated Digital LAMP Validation Using SARS-CoV-2 Orf1ab Visual RT-LAMP Kits

Pseudoviral plasmid reference material and negative control derived from the SARS-CoV-2 Orf1ab Visual RT-LAMP Kit (contained 10^6^ copies/μL = 10^9^ copies/mL, Beijing, China) was tested via the simulated digital LAMP used in the present study.

The target concentration was prepared at a 2-step concentration gradient (10^4^, 10^3^ copies/mL) and additionally verified via qPCR by Zoonbio Biotechnology Co. Ltd (Nanjing, Jiangsu, China). based upon directions provided with the SARS-CoV-2 Orf1ab Visual RT-LAMP kit.

### 3.8. The Specificity and Assessments of the Simulated Digital LAMP

To determine the specificity of the developed method in this paper, contrived nasopharyngeal (NP) swabs (*n* = 18) and 2 common human coronavirus samples (strain 0C43, NL63), inactivated MERS-CoV and SARS-CoV-1 set as negative control were directly tested by simulated digital RT-LAMP assay and RT-qPCR to evaluate the specificity of the test. RT-qPCR were tested by the hospital of Nanjing.

## 4. Conclusions

Herein, through electrophoretogram of RT-LAMP for the detection of SARS-CoV-2 viral RNA, we found the dATPαS reacted well in RT-LAMP, which could take place of dATP. Then by coupled RT-LAMP with digital PCR, our digital LAMP system (Rainsuit, Suzhou, Jiangsu, China) with our optimized RT-LAMP-BART was able to achieve absolute quantification while conventional digital LAMP system (Rainsuit) was not. However, the digital LAMP system (Rainsuit) could not detect viral RNA in real-time and had a detection limit of 10^4^ copies/mL owing to the limited use of sample copies analyzed in this reaction. The sensitivity of our simulated digital LAMP-BART approach was as low as 10 copies/mL, making this assay far more sensitive than digital LAMP and can be detected in real-time, yielding results within 20 min. This simulated digital LAMP assay was also easier to conduct relative to digital LAMP as it did not require preparation of droplet generation or stabilizers. The sensitivity and stability of this simulated digital LAMP assay can also be further improved by increasing the reaction volume and the total number of wells used in a given analysis.

As such, the simulated digital LAMP approach developed in the present study is highly sensitive, stable, and efficient, offering a means of reliably quantifying the absolute amount of target viral RNA present in a given sample of interest.

## Figures and Tables

**Figure 1 ijms-22-01017-f001:**
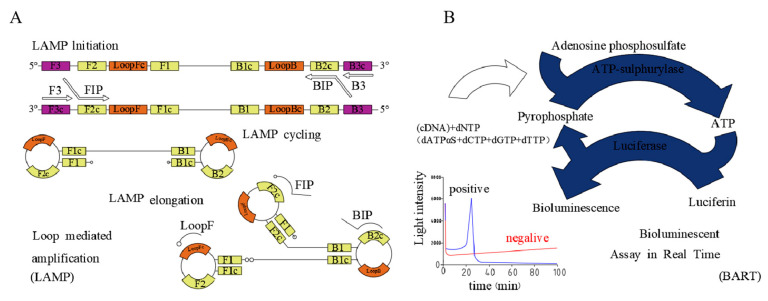
RT-LAMP-BART and LAMP assay principles. (**A**) RT-BART schematic. (**B**) LAMP schematic.

**Figure 2 ijms-22-01017-f002:**
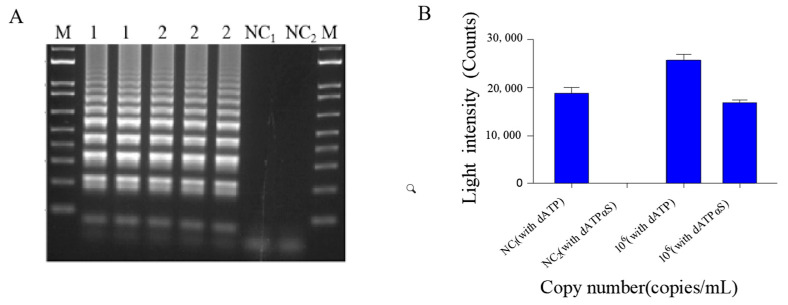
(**A**) Agarose gel electrophoresis of RT-LAMP products. Lanes NC_1_, 1, 2, and M correspond to the negative control, 10^6^ copies/μL of the purified template pseudoviral plasmid reference RNA (with dATP), 10^6^ copies/μL of the purified template pseudoviral plasmid reference RNA (with dATPαS) and the marker (DL2000), respectively. (**B**) Light intensity (counts) of the purified template pseudoviral plasmid reference RNA (with dATP), 10^6^ copies/μL of the purified template pseudoviral plasmid reference RNA (with dATPαS) and the negative control of dATP and dATPαS.

**Figure 3 ijms-22-01017-f003:**
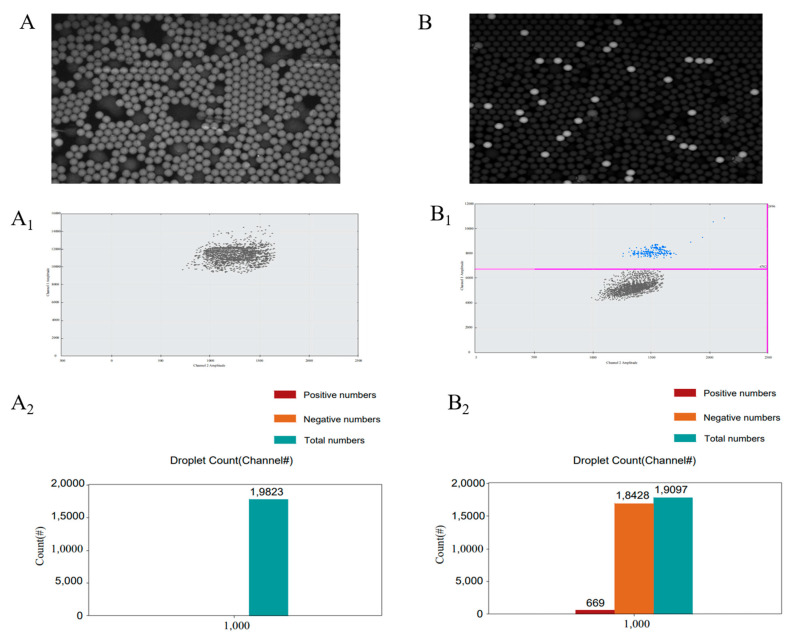
(**A**) RT-LAMP-BART bioluminescent droplet images prepared using a conventional BART reagent mixture [17]. (**A_1_**) The distribution of bioluminescent counts in (**A**). (**A_2_**) Droplet count classifications in (**A**). (**B**) RT-LAMP-BART bioluminescent droplet images prepared using an optimized BART reagent mixture containing dATPαS rather than dATP. The analyzed sample was the same as in (**A**). (**B_1_**) The distribution of bioluminescent counts in (**B**). (**B_2_**) Droplet count classifications in (**B**).

**Figure 4 ijms-22-01017-f004:**
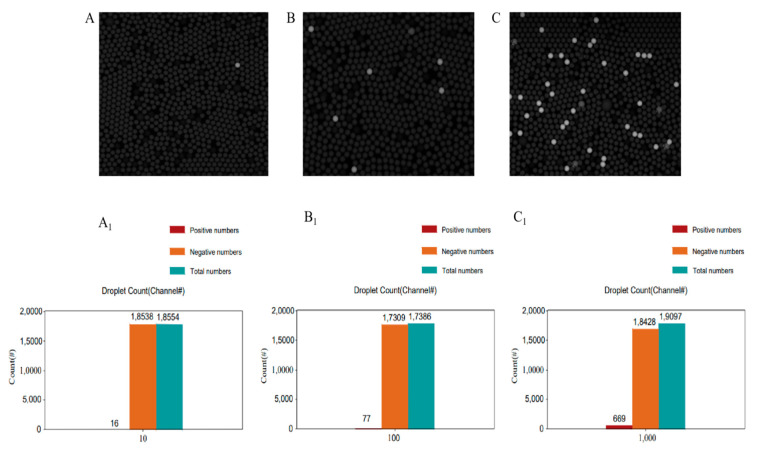
(**A**) RT-LAMP-BART bioluminescent droplet images prepared using an optimized BART reagent mixture. The target RNA template concentration was 10^4^ copies/mL. (**A_1_**) Droplet count of the reaction shown in (**A**). Positive droplets: 9; negative droplets: 19,574; total droplets: 19,583. (**B**) RT-LAMP-BART bioluminescent droplet images prepared using an optimized BART reagent mixture. The target RNA template concentration was 10^5^ copies/mL. (**B_1_**) Droplet count of the reaction shown in (**B**). Positive droplets: 77; negative droplets: 17,309; total droplets: 17,386. (**C**) RT-LAMP-BART bioluminescent droplet images prepared using an optimized BART reagent mixture. The target RNA template concentration was 10^6^ copies/mL. (**C_1_**) Droplet count of the reaction shown in (**C**). Positive droplets: 669; negative droplets: 18,428; total droplets: 19,097.

**Figure 5 ijms-22-01017-f005:**
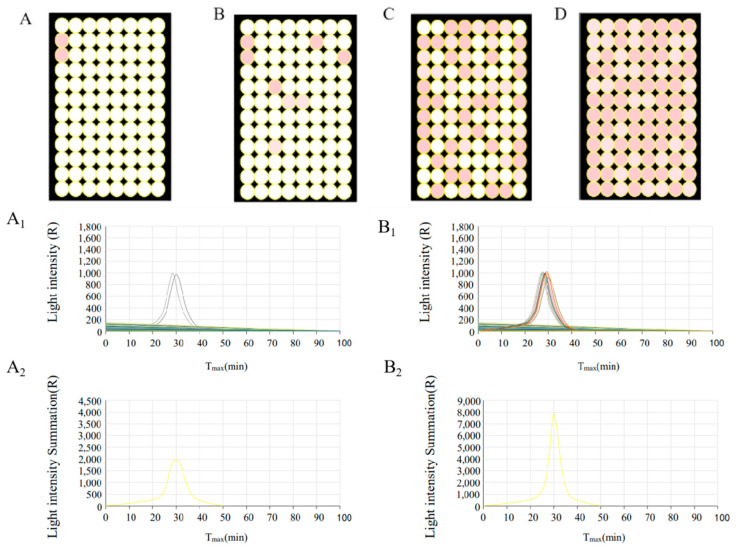
(**A**) RT-LAMP-BART bioluminescent well images obtained when utilizing an optimized BART reagent mixture. The target RNA template concentration for this plate was 10 copies/mL. Positive wells: 2 Negative wells: 94 Total wells: 96. (**B**) RT-LAMP-BART bioluminescent well images obtained when utilizing an optimized BART reagent mixture. The target RNA template concentration for this plate was 10^2^ copies/mL. Positive wells: 8; negative wells: 88; total wells: 96. (**C**) RT-LAMP-BART bioluminescent well images obtained when utilizing an optimized BART reagent mixture. The target RNA template concentration for this plate was 10^3^ copies/mL. Positive wells: 58; negative wells: 38; total wells: 96. (**D**) RT-LAMP-BART bioluminescent well images obtained when utilizing an optimized BART reagent mixture. The target RNA template concentration for this plate was 10^4^ copies/mL. Positive wells: 96 Negative wells: 0 Total wells: 96. (**A_1_**) RT-LAMP-BART amplification curve of each well in (**A**). (**A_2_**) The summed RT-LAMP-BART amplification curve for all wells in (**A**). (**B****_1_**) RT-LAMP-BART amplification curve of each well in (B). (**B_2_**) The summed RT-LAMP-BART amplification curve for all wells in (**B**).

**Figure 6 ijms-22-01017-f006:**
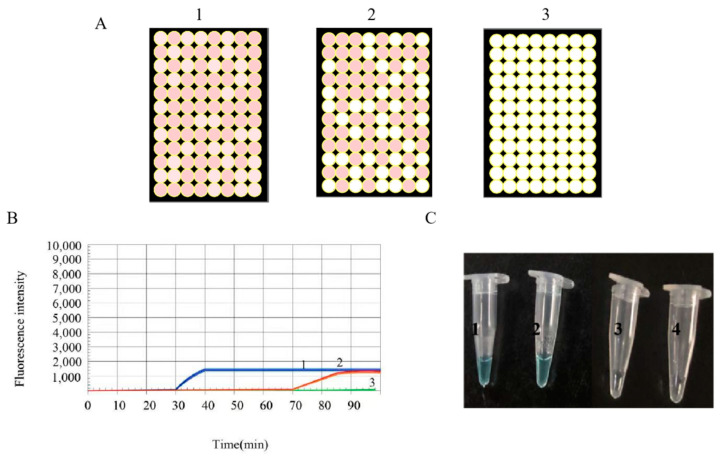
(**A**) RT-LAMP-BART bioluminescent well images obtained when utilizing an optimized BART reagent mixture. 1 and 2 meant the target RNA template concentration from 10^4^ to 10^3^ copies/mL. 3 meant the negative control sample. Positive wells from 1 to 3: 96, 60, 0. Negative wells from 1 to 3: 0, 30, 96. The total well wells from 1 to 3: 96, 96, 96. (**B**) The same samples in (**A**) were tested by qPCR. (**C**) The same samples in (**A**) were tested via a colorimetric kit, wherein blue and clear correspond to positive and negative, respectively. The negative control sample was derived from the SARS-CoV-2 Orf1ab Visual RT-LAMP Kit.

**Table 1 ijms-22-01017-t001:** The accuracy and sensitivity of a bioluminescent signal-based digital LAMP assay.

Target RNA Concentrations (Copies/mL)	Target RNA Concentrations in Reaction Mix (Copies/μL) (10 μL Reaction Mixtures)	Theoretical Calculation (N = 20,000)			Actual Results			
		λ	Np	Nn/N	Np	N	λ	C (Copies/μL)
10^4^	1	0.0005	10	0.9995	9	18,554	0.00049	0.98
10^5^	10	0.0050	99	0.9950	77	17,386	0.00443	8.89
10^6^	100	0.0500	969	0.9516	669	19,097	0.03503	70.06

N: the total number of droplets. Np: number of positive droplets. Nn: number of negative droplets. λ: copies per droplet = −ln(Nn/N) = Copies in the reaction/Total droplets = Concentration in reaction (Copies/μL) × V (volume of each droplet). V = Reaction volume/N = 0.0005 μL. P(0) = Nn/N = e^−λ,^ λ = −ln(Nn/N) = Concentration in reaction × V. Concentration in reaction = [−ln(N/N)]/V. Target RNA concentration = 10 × C_Copies/μL_ = 10^43^ × C_Copies/mL_. P(0) = Nn/N = e^−λ^, λ = −ln(Nn/N) = C_Copies/mL_ × V.

**Table 2 ijms-22-01017-t002:** The accuracy of the bioluminescent signal-based simulated digital LAMP assay.

Target RNA Concentrations (Copies/mL)	Target RNA Concentrations in Reaction (Copies/μL) (1000 μL Reaction Mixtures)	Theoretical Calculation (N = 96)			Actual Results			
		λ	Np	Nn/N	Np	N	λ	C (Copies/μL)
10	0.001	0.0104	1	0.9896	2	96	0.0208	0.0019
10^2^	0.01	0.1040	10	0.9019	8	96	0.0870	0.0084
10^3^	0.1	1.0416	62	0.3535	58	96	0.9268	0.0885
10^4^	1	10.4167	96	0	96	96	≥9.6	≥0.96

N: the total number of wells. Np: number of positive wells. Nn: number of negative wells. λ: copies per well = −ln(Nn/N) = Copies in the reaction/Total wells = Concentration in reaction (Copies/μL) × V (volume of each well). V = Reaction volume/N = 10.4 μL. P(0) = Nn/N = e^−λ^, λ = −ln(Nn/N) = Concentration in reaction × V. Concentration in reaction = [−ln(N/N)]/V. Target RNA concentration = 10 × C_Copies/μL_ = 10^4^ × C_Copies/mL_.

**Table 3 ijms-22-01017-t003:** The accuracy of the bioluminescent signal-based simulated digital LAMP assay.

Target RNA Concentrations (Copies/mL)	Target RNA Concentrations in Reaction (Copies/μL) (1000 μL Reaction Mixtures)	Theoretical Calculation (N = 96)			Actual Results			
		λ	Np	Nn/N	Np	N	λ	C (Copies/μL)
0	0	0	0	1	0	96	0	0
10^3^	0.1	1.0416	62	0.3535	60	96	0.9808	0.0943
10^4^	1	10.4166	96	0	96	96	≥9.6	≥9.6

N: the total number of wells. Np: number of positive wells. Nn: number of negative wells. λ: copies per well = −ln(Nn/N) = Copies in the reaction/Total wells = Concentration in reaction (Copies/μL) × V (volume of each well). V = Reaction volume/N = 10.4 μL. P(0) = Nn/N = e^−λ^, λ = −ln(Nn/N) = Concentration in reaction × V. Concentration in reaction = [−ln(N/N)]/V. Target RNA concentration = 10 × C_Copies/μL_ = 10^4^ × C_Copies/mL._

**Table 4 ijms-22-01017-t004:** Ct of qPCR products produced from different copies of standard plasmid pMD18T-S.

Methods	Results		
	Positive	Negative	Minutes for Positive Call
qPCR	9	11	48 min
simulated digital PCR	9	11	30 min

**Table 5 ijms-22-01017-t005:** LAMP assay primer designs.

Primer	Primer Sequence (5′-3′)	Target Gene
FIP	CACAACTACCACCCACTTTTGCCATGCAAGTTGAATC	ORF1ab
BIP	CGGACACAATCTTGCTAATAAGAAGTTGAATGTCTTCACC	
F3	AACATGGAGGAGGTGTTG	
B3	CAAGTAGAACTTCGTGCTG	
LoopF	GTGGTCCATTAGTAGCTATGT	
LoopB	CACTGTCTTCATGTTGTCG	

## Data Availability

Not applicable.

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
