# Peer review of "A Novel Approach to the Bioluminescent Detection of the SARS-CoV-2 ORF1ab Gene by Coupling Isothermal RNA Reverse Transcription Amplification with a Digital PCR Approach"

_ijms, 2021, doi:10.3390/ijms22031017_

Round 1
Reviewer 1 Report
This is a technical report of a novel testing of SARS-CoV-2 detection by combining LAMP and degital PCR approach. This is an interesting paper but there are several issues to improve manuscript.
- It seems A and B are opposite in Fig 1.
- Tables. It is hard to see which are" theoretical calculation" and "actual results". Please separate these by space or vetical line.
- Rational of a combination of LAMP and PCR is unclear and should describe more easily for broad range of readers. Demerit of this apporoach should also be discussed. Time required for this testing should be mentioned. Cost issues should be mentioned.
Author Response
First of all . I want to thank reviewer 1 for the comments of reviewer 1. It helped me a lot.
1: A and B are not opposite in Figure 1. Figure 1A showed the principle of LAMP amplification, and Figure 1B showed the principle of bioluminescence works. Maybe due to the Figure 1 was not clear enough, I have changed to a higher definition Figure 1.
2: The theoretical results and the actual results in the tables have been separated.
The format of the tables been modified according to the requirements of the journal
3: This manuscript is an attempt and optimization of the combination of LAMP-BART and digital PCR, and a discussion of time and cost has been added in Page 10, Line300-310. Section 2.5.
Reviewer 2 Report
Dear Authors,
After review of your manuscript I suggest the following major points to be adressed:
- The specificity of your assay have to be discussed. This is a critical issue for the performance of your assay ion real life. Please argument by using alignemnts od DNA sequences that your assay is specific.
- A figure showing the primers position is in my point of view necessary.
- The assay while nicely validated in vitro should be tested in screening of positive samples. This is critical to assesss the specificity of the assay, to test for false and negative positives and to determine overall performance of the assay in real laboratory screening.
Minor points :
- The quality of figure 1: your figure have to be high resolution. This is mandatory for the efficient understanding of this figure.
Author Response
First of all . I want to thank reviewer 2 for the comments of reviewer 2. It helped me a lot.
Major points
1: 3: There are no clinical trials in this manuscript because the clinical samples of SARS-CoV-2 in hospital are strictly controlled. We didn't have a sample yet,The Results and Discussion section 3.8 in the manuscript were added to iillustrate the specificity of this method. Due to time constraints, I can only use a small number of samples to prove the specificity of this method, hoping you to understand. The idea of this manuscript lies in the development of a new method combining LAMP-BART with digital PCR, which can be applied to the detection of various pathogenic microorganisms
2:The figure showed that the position of primers cannot be provided at present, because it is in the confidential stage, please understand. Thank you.
Minor points :
I have changed to a higher definition Figure 1.
Round 2
Reviewer 2 Report
Dear Authors,
I understand your arguments for my requests. Please look carefully at these points. I can now suggest acceptance of the paper to promote scientific discussions and progress in this important field.
Best regards
Author Response
First of all . I want to thank reviewer 2 for the comments of reviewer 2. It helped me a lot. Reviewer 2 is very rigorous
The small deficiencies have been revised by my tutor again.